# Health Risks Management Program in Schools: An Operational Study in Türkiye

**DOI:** 10.3390/ijerph20043718

**Published:** 2023-02-20

**Authors:** Muhammed Atak, Ayşe Emel Önal, Zeynep Şimşek, Halim İşsever

**Affiliations:** 1Department of Public Health, Istanbul Medical Faculty, Istanbul University, 34093 Istanbul, Türkiye; 2Faculty of Health Sciences, İstanbul Bilgi University, 34060 Istanbul, Türkiye

**Keywords:** school health, children and adolescent, management of health risks, intervention, operational epidemiology

## Abstract

This study was carried out to improve the quality of school health services with the operational epidemiology method. The study aimed to describe the current status of the School Health Protection and Improvement Program (SHPIP), the difficulties experienced during its implementation, to develop evidence-based solution methods, and to test the proposed solutions, in a district with a population of 400,513, 20.4% of which is of school age between the ages of 5–19. The “Health Risk Management Program at Schools”, which consists of the stages of putting the results into practice by sharing the results with the relevant parties, wasdeveloped. In this study, a cross-sectional research design was employed using questionnaire forms, while qualitative data were collected through the use of the phenomenological analysis method, specifically through the conduct of focus group interviews. SHPIP year-end evaluation forms of 191 schools were reviewed retrospectively, questionnaires were applied to 554 school staff and 146 family health center staff between 21 October 2019 and 21 November 2019 using the probabilistic sampling (simple random) method, and semi-structured focus group interviews were conducted with 10 school health study executives. The most common health risks in schools and the risks identified during the execution of school health services were identified. In order to eliminate the lack of in-service training, training modules for “School Health Management Teams” were developed and impact analyses were conducted. After the intervention, a significant difference was found in the compliance of schools with SHPIP, and the application of all components of the school health program increased from 10.0% to 65.6% (*p* < 0.05). The program has been integrated into the “School Health Protection and Improvement Program” (SHPIP) with the decisions of the District School Health Board and District Hygiene Council.

## 1. Introduction

Since childhood is the main determinant of healthy life years in adulthood and old age, it is important to prevent, and provide early diagnosis and treatment of diseases and reintegration into society [1]. For this reason, school health studies are of critical importance in this period when school growth and development is the fastest [2,3].

School health is defined as all the activities carried out to evaluate the health of school staff as well as students, to support and develop the physical, mental and social development of students, to provide and maintain a healthy school environment, and to provide health education to students and indirectly to society [4]. School health services cover many topics for students and school staff as follows: the planning and implementation of programs and health education for the protection and development of health, the prevention of possible health risks and problems and management of existing health problems, making necessary follow-ups about health and keeping records, and the retention and evaluation of health problems and services. In this context, risky health behaviors in school-age children should be determined and appropriate health activities and studies should be carried out. Risky health behaviors in school-age children are smoking, alcohol and substance consumption, unhealthy diet, sedentary life, unwanted accidents, violence and risky sexual behaviors. Adolescence is a period that is prone to acquiring risky behaviors, and risky behaviors gained in this period are among the factors that reduce the quality of life in adulthood by reflecting on adulthood [5,6,7,8,9,10]. The school age range encompasses a wide spectrum, from preschool and primary school-aged children to late adolescents and young adults. The health needs and behaviors of children in this range vary greatly, and these differences should be taken into account when planning school health services. For this reason, a tailored approach that takes into account the different age ranges should be adopted when defining school health services.

Depending on the reasons explained above, school health programs have been developed to protect and improve the physical, mental and social well-being of children who form the future of society. In the 19th century, school health programs started to be implemented in the world, and in the 20th century, school health services were included in the scope of a routine service in developed countries [11].

In Türkiye, the first legal regulations regarding school health started with the Public Health Law in the 1930s, and it was stated that school health services were under the obligation of the state [12].

Various studies have been carried out in the field of school health in Türkiye [13,14,15,16]. The “School Health Cooperation Protocol” signed between the Ministry of National Education and the Ministry of Health was revised and signed and the “School Health Protection and Improvement Program (SHPIP)” was initiated. With this program, all programs and projects carried out or to be carried out in schools were handled with a comprehensive and holistic approach. Within the scope of this program, six study areas were determined as “health services, healthy and safe school environment, healthy nutrition, health education, physical activity, family/community participation”. With the follow-up charts prepared, it was aimed to monitor and evaluate these areas by the Ministry of National Education and the provincial organizations of the Ministry of Health [17]. According to 2019–2020 data, a total of 18,241,881 students receive education at preschool, primary and high school levels in Türkiye [18]. If teachers and school staff are added to this number, we can state that school health studies concern a significant part of the society.

In the studies reported in the literature, attention is drawn to the problems that can be experienced in the field applications of school health programs [19,20]. Despite the legal and structural regulations explained above, it is stated that there are deficiencies in the implementation of programs related to school health except immunization [21]. School Health Evaluation Teams visit the schools in each school term and evaluate the schools on the forms and data determined in line with the SHPIP guideline. As a result of the data obtained after these evaluations, and the request of some school administrators who encounter health risks in the school environment for help in managing these risks, this area has been focused on and the search for solutions has been started. 

School health research and evaluation studies can help improve school health programs and enhance the implementation of methods applied in different countries around the world. This is because they provide a better understanding of the health status and health needs of students in schools, which can lead to the development of more effective and efficient school health programs. Evaluating practices from different countries can also offer new ideas and approaches for addressing similar health problems. In this way, school health research and evaluation studies can play a role in helping students achieve a healthy and successful educational experience. From an international perspective, it is important to recognize the value of school health programs in ensuring the well-being and success of students globally. By sharing best practices and conducting cross-cultural evaluations, we can work towards creating a more equitable and effective system of school health care for all students, regardless of their location or background.

This study aimed to evaluate the SHPIP, which is in force in our country, and to prepare a program that can support it by determining its failing aspects. This study is the first epidemiological study conducted to improve the quality of school health programs in Türkiye and provided important guiding information for other studies. 

## 2. Methods

This operational study was carried out by Eyupsultan District Health Directorate with the cooperation of Eyupsultan District Governorship, Eyupsultan National Education Directorate and other relevant public institutions. Operational research involves defining the current problem and any challenges encountered in its implementation, developing evidence-based solutions, testing the proposed methods and finally sharing the results with relevant stakeholders and putting them into action [22]. 

### 2.1. Descriptive Features of the Study Area

Eyupsultan district, located in the province of Istanbul in Türkiye, is a large settlement area of 242 km^2^ [23]. According to Turkish Statistical Institute (TURKSTAT) 2019 data, 400,513 people live in the district, and school-age children between the ages of 5–19 constitute 20.4% of the population. In the 2018–2019 academic year, there were 191 schools in Eyupsultan district, including 63 preschools, 43 primary schools, 46 secondary schools and 39 high schools. The number of teachers was 3285.

### 2.2. Operational Study Design

Research permission for the studies to be carried out in the district was obtained from Eyupsultan District Governorship on 18 July 2019, ethics committee approval with file number 2019/956 from Istanbul Medical Faculty Clinical Research Ethics Committee on 9 August 2019 and institutional permission from Istanbul Provincial Health Directorate on 20 September 2019.

### 2.3. Determining the Current Status of the SHPIP

A retrospective analysis of the follow-up forms specified in the School Health Program Implementation Guide, which was obtained during the inspections of the District Health Directorate and District National Education Directorate employees in the 2017–2018, 2018–2019 academic years, in the scope of the “School Health Protection and Improvement Program” in Eyupsultan district was conducted. It has been determined which of the SHPIP components were fulfilled by the schools and to what extent. While the schools were being inspected, the schools were also evaluated in terms of cleanliness and hygiene within the scope of the “White Flag Project”. The preparation levels of the schools related to the “Nutrition-Friendly School Project”, which aims to carry out studies on healthy nutrition and active living, were also examined. It was determined that 105 (61.7%) schools out of 170 schools in the 2017–2018 academic year did not comply with SHPIP at all, 65 schools partially complied and 9 schools completely implemented the program (Table 1).

In the follow-up forms, there were 48 questions about health services, healthy and safe school environment and healthy nutrition. In the 2018–2019 academic year, the current situation associated with the health risks in the school during the inspections at schools is given in Table 2.

The method to be followed using the community organization method was determined by interviewing the District Governor’s Office, District Health Directorate, District National Education Directorate, District Municipality, Youth and Sports Directorate, Agriculture, Food and Livestock Directorate, District Police Department and University representatives in the District Health Board. 

In this context, these decisions were taken: ➢All schools were included in the scope of the study;➢Using the cross-sectional research method and qualitative research methods, to identify the problems and solution proposals that emerged during the implementation of the program reported by the school administrators and teachers responsible for the execution of the OSKGP, and the employees of the Family Health Center and the District Health Directorate;➢Development and implementation of the intervention program based on research findings;➢Strengthen the cooperation between health service providers, school staff and public institutions.

### 2.4. Development of Data Collection Tools

Questionnaire for school staff: A questionnaire was prepared based on the SHI School Health Index—A Self-Assessment and Planning Guide published in 2017 by the United States (USA) Center for Disease Control and Prevention, Department of Health and Human Services and SHPIP implementation guide [24,25]. The reliability and validity of the School Health Index (SHI) have been widely studied and generally found to be satisfactory. The SHI has been used for many years and has undergone numerous revisions and updates based on feedback from users and research findings. Reliability refers to the consistency or stability of a measure over time. The SHI has been found to be reliable, meaning that it consistently provides similar results when used by different users or when used at different times. This is important because it means that schools can use the SHI to accurately assess their health policies and programs and track their progress over time. Validity refers to the accuracy of a measure in reflecting what it is intended to measure. The SHI has been found to be valid, meaning that it accurately measures the aspects of school health policies and programs that it is intended to measure. This is important because it means that schools can use the SHI to make meaningful improvements to their health policies and programs and to assess the impact of those improvements on student health and academic success. Overall, the School Health Index (SHI) is considered to be a reliable and valid tool for assessing school health policies and programs, and has been widely used by schools, health departments and other organizations for many years.

A total of 81 questions were included in the survey, including the sociodemographic characteristics of the participant (12 questions) and school environment, health education, physical activity, nutrition, employee health and parent–school cooperation. After the pilot study of the survey was conducted, it was aimed to reach 345 people out of 3285 school employees in total, according to the sampling calculation made, and 554 people answered our survey. 

Questionnaire for Family Health Center employees: There were 25 questions in total, 8 of which were demographic information and 17 of which included information about school health, evaluating the responsibilities and awareness of family physicians determined by legal regulations in the school health program. After the pilot study of the questionnaire was completed, 146 people (78.1%) completed the questionnaire, which was sent electronically to 187 healthcare professionals.

Focus group interview with the people responsible for the execution of the school health program: A semi-structured focus group interview was conducted with a total of 10 people, 5 employees from the District Health Directorates responsible for overseeing school health services and 5 employees from the District National Education Directorates, using the purposeful non-probability sampling method. The semi-structured questionnaire included questions on demographic information, questions regarding the school health services program, school health risks and suggestions for improving the system as the executors of the program. The focus group interviews lasted approximately three hours and were recorded using a sound recording device. The recordings were later transcribed and subjected to content analysis, which was performed manually without the use of any software. All interviews were conducted by the principal researcher (MA). The researcher had no relationship with participants before, during or after the study. The participants were informed about the study, and the researcher introduced herself and explained the aim of the study and the data collection (interview) procedure. All interviews were conducted in the District Health Directorate. The researcher took field notes during interviews if needed. All interviews were audio-recorded with the participants’ permission. After the transcripts were obtained, they were given to the participants for review. Additionally, assessments were made using the Consolidated Criteria for Reporting Qualitative Research (COREQ) checklist by Tong et al. (2007).

### 2.5. Data Analysis

SPSS 22.0 program (IBM, Armonk, NY, USA) was used in data entry and analysis, and intergroup comparisons were made based on 95% confidence level as well as descriptive statistics. Microsoft Word^®^ was used in analysis of focus group interviews.

## 3. Results

### 3.1. Results Regarding the Determination of the Intervention Method

#### Results Obtained from School Staff

In total, 82.5% of the respondents were teachers, 9.6% were deputy principals, 6.3% were principals and 1.6% were other personnel. A total of 66.6% of our participants were male and 33.4% were female, and the mean age was 37.17 ± 8.98 years. In total, 87% of 554 school employees worked in public schools, and 13% work in private schools. Most of the participants (37.0%) worked in high school, followed by those working in secondary school (28.3%), primary school (28.0%) and preschool (3.8%). A total of 70.4% of the participants reported that they had worked at the same school for 5 years or less. In total, 84.8% of the participants reported that they had not received any training on school health before. Responses of the participants regarding health risks and their management are given in Table 3. 

As seen in Table 3: ➢A total of 59.0% of the participants stated that there was no school health management team plan and related written documents and guides;➢Practices involving risk management for school accidents, emergencies and violence control were not carried out at rates ranging from 25–76%;➢What needs to happen to prevent tobacco, alcohol and substance use was not carried out at rates ranging from 40.6 to 60.3%;➢The proportion of school employees who had a first aid certificate showing their knowledge and skills of responding to accidents and injuries was 24.0% of all employees;➢In total, 80.6% of the schools were not working to effectively fight against the risks of contagious diseases;➢Risk management practices related to environmental health were not implemented in schools at rates ranging from 31.4% to 70.0%;➢Good practices regarding the prevention of unhealthy nutrition and sedentary life in schools varied between 28.4% and 74.1%; ➢Studies on the prevention of mental health and behavioral problems in schools were not carried out at rates ranging from 26.1% to 47.3%.

### 3.2. Results Obtained from FHC Staff

In total, 75.3% of the 146 FHC staff participating in the study were women, and the mean age of the participants was 38.0 ± 7.8 years. As a result of the analysis, the following were determined:➢In total, 80.1% of them reported that they did not receive any training within the scope of school health services.➢The rate of knowing the content of health programs and projects carried out in schools by FHC staff varied between 17.0% and 91.0%.➢It was seen that the least informed programs were the Diabetes Education Program at Schools (17.8%), the White Flag Project (21.2%) and Studies for the Prevention of Violence in Schools (24.0%).➢It was stated that the most common problems in children that applied to FHCs were endocrine, nutritional and metabolic diseases (30.0%), respiratory system diseases (15.7%) and infectious diseases (9.3%).➢Within the scope of child and adolescent follow-up parameters, 21.2% of the participants reported that they were involved in school-age children’s weight, body mass index, hearing, vision, developmental assessment, social behavior assessment, physical examination, vaccination/immunization, hyperlipidemia risk assessment, oral–dental health, counseling and hemoglobin/hematocrit follow-up.➢In total, 78.1%. of FHC employees reported that they did not contact school administrations.

### 3.3. Results Obtained from Focus Group Discussions

The aim of the focus group interviews conducted with these authorized individuals was to identify issues related to school health. In this context, various questions were asked and main themes related to the problems were shaped. The question “What did school health mean to you before you took charge of school health?” was asked. Five of the participants emphasized the themes of “canteen”, four of them “vaccination” and two of them “school cleaning”. When asked about what school health risks meant to this group responsible for school health assessments, six people emphasized the themes of “Communicable diseases and infections”, and five people emphasized the themes of “hygiene-related risks”, “unhealthy diet/obesity” and “school accidents”.

### 3.4. Findings-Based Intervention Programs

Considering the results of record reviews, survey and focus group discussions, the following interventions were identified and implemented within the scope of SHPIP:➢Ensuring the establishment of ‘School Health Management Teams’ in all schools with the decision taken by the District School Health Board (school administrator, a teacher, counselors, a student, a member of the parent–teacher association) and holding information meetings,➢Preparing school health awareness trainings within the framework of risk management, lasting 4 h on average, based on health education theories to ensure behavior change, and providing trainings by 4 different teams formed from the District Health Directorate in an integrated manner with teachers’ routine training seminars,➢Informing FHC staff about the routine follow-ups of school-age children and conducting follow-up visits,➢All interventions are decided by the District School Health Board.

Risk-management-focused “Certified First Aid”, “Communicable Diseases”, “Hygiene and Environmental Health”, “Healthy Nutrition and Physical Activity”, “Mental and Behavioral Problem Management” and “Addiction and Substance Use Awareness” training modules were prepared. In these training modules, the rule of “Diagnose, Prevent, Intervene, Direct” (DPID) was adopted. In line with this defined DPID rule, it was intended for individuals to be able to recognize health risks at school, prevent and intervene in these risks in line with the information they receive when they encounter these health risks, and make use of the necessary guidance regarding the said health risk with the developed institutional cooperation and communication network. Pretest and posttests were prepared for each module and applied before and after the training. 

### 3.5. Results of Intervention Program

Six months after the intervention, all schools were visited and evaluation forms within the scope of SHPIP were completed, and semi-structured focus group interviews were conducted with six randomly selected people from the school health management teams for impact monitoring.

➢Evaluation of Training Programs for School Staff✓First Aid Training

During January 2020, first aid training was carried out with 15 different groups. A total of 304 people attended the training in groups of 21 people. Each group training lasted 2 days and each group was given 16 h of training in total. The training sessions were held simultaneously with more than one group in different centers between 30 December 2019 and 9 January 2020. After the training, written and practical exams were held, and the pass score was determined as 85.

Within the scope of the training module implemented, a first aid exam was administered to those in the School Health Management Team. Accordingly, a first aid exam was administered to a total of 304 personnel, including 295 teachers and school administration personnel, and 9 personnel from the Directorate of National Education. While 69.1% of those who take the exam were women, 76.3% worked in the public sector and 63.6% worked as teachers. While 93.7% of the participants passed the first aid exam, 19 participants (6.3%) failed The results of the exam after the first aid training are shown in Table 4.

✓Other Trainings and Pre–Posttests

On 15 January 2020, the training module on the Management of Health Risks at School for school employees was held in two groups. The trainers were selected from the doctors working in the District Health Directorate. The training was given to 156 people in total, 4 h for each group. Pre and posttests were applied before and after the five training modules, and the number of people who completed these tests was 117. As seen in Table 5 and Table 6, it was determined that the pretest–posttest mean scores differed significantly (*p* < 0.05).

✓Evaluation of Training Programs for Family Health Center Staff

In March 2020, the training module on the Management of Health Risks at School for family health center employees was held in two sessions for 2 h. In the content of the training, the responsibilities of SHPIP and FHC employees within the scope of this program were explained. The trainers were selected from the doctors working in the District Health Directorate and responsible for the unit related to the subject. In total, 89 family physicians and 88 family health workers attended the training.

✓Impact Assessment Based on SHPIP Assessment Forms

When the 160 schools that were visited in the 2018–2019 academic year and the 2019–2020 academic year were examined, significant improvements were found in the Health Service component, the Healthy and Safe School Environment component and the Healthy Nutrition component, as seen in Table 7 (*p* < 0.001). In addition to these components, the proportion of schools with Nutrition Friendly and White Flag certificates had increased significantly over the years (*p* < 0.001).

✓Integration and Dissemination of the Prepared Program into the System

An intervention study was conducted with the “Health Risk Management Program in Schools”, which was prepared for the health risks and missing information identified after the needs analysis. The “Health Risk Management Program in Schools”, which was prepared by the decision of the District School Health Board and the decision of the District Hygiene Council, was integrated into the school health services program of the district. A detailed report on this program was shared with the Provincial Health Directorate and Provincial Directorates of National Education, and it was recommended to disseminate it in other districts.

## 4. Discussion

In this study, the “Health Risk Management Program in Schools” developed with the operational research method to increase the effectiveness of SHPIP and the process of integrating the program into the system were explained. Kazemitabar et al. reported the inadequacy of studies on the evaluation of school health programs [26].

The initial stage of the study utilized both quantitative and qualitative research methods to determine the current state of SHPIP and the awareness and understanding of service providers. For the quantitative design of the study, survey forms were used, and for the qualitative design, focus group interviews were conducted. Within the scope of the study, the level of knowledge was determined with the CDC’s School Health Index, which provides the opportunity for schools to self-evaluate and is used in many countries [24,27,28]. In total, 85% of the school staff stated that they had not received any training on school health. Similar to the results of this study, it was determined in the literature that the level of knowledge among school staff was low [29,30,31]. This finding shows that a teacher education program on school health is a necessity before and after graduation [21]. In other studies, it has been shown that teachers’ school health education is insufficient, they mostly obtain information from internet resources, similar to the findings of the study, and in-service training programs are not implemented [32,33,34,35,36,37].

We determined that three-quarters of the participants did not have a first aid certificate. It is noteworthy that the number of participants with first aid certificates was quite low, and similar rates were also found in the literature. In a study, 68.4% of teachers stated that they had not previously received first aid training [38,39].

It was determined by the school staff that the most common health problems in students were respiratory system diseases, followed by digestive system diseases (nausea–vomiting, abdominal pain, diarrhea, constipation, etc.), and infectious and parasitic diseases. In the Queensland region of Australia, the most common health problems in school-age children over the age of 5 were reported as upper respiratory tract diseases, gastrointestinal system diseases, allergies, infections and parasitic diseases, respectively [40]. In a study conducted in Uganda, malaria, diarrhea, skin fungal infections, influenza and typhoid fever were among the most common diseases in children [41]. In the report published by TURKSTAT in June 2020, upper respiratory tract infection ranked first with 29.4% in the school-age population, followed by diarrhea with 18.3% and oral and dental health problems with 14.2% [42].

In our study, it was determined that the preparation of an emergency health response plan for injuries, anaphylaxis, asthma attacks, allergies, etc., the information provided to all personnel on these issues and the follow-up of the health status of students with acute and chronic diseases were found to be insufficient. Similar results were reported in other conducted studies [43,44]. It was evaluated that it would be beneficial to inform the school staff about the approach to emergency health situations and students with acute and chronic diseases and to prepare an action plan in this direction.

In the second step of the operational research, the “Health Risks Management Program in Schools” was developed for school staff to gain the knowledge and skills to manage health risks that may be encountered in schools.

It was determined that the majority of FHC staff, similar to school staff, up to four-fifths, had not received any training on school health services. A similar rate was found in another study [35]. The fact that FHC staff, who are primarily responsible for the execution of school health services, do not have knowledge about the school health program, results in the failure to fulfill some of their duties. For this reason, training and awareness activities were carried out for FHC employees within the scope of SHPIP.

When the evaluation forms obtained from the inspections carried out within the scope of OSKGP in schools were examined, it was seen that the level of meeting the criteria in the fields of Health Service, Healthy and Safe School Environment, and Healthy Nutrition increased significantly. When these three components are evaluated according to years, the rate of completeness, which was 17.65% in 2017–2018 for studies in the health care title, increased to 31.55% in the following year and finally to 71.91% in 2019–2020 after the training module. In other words, the number of schools with a full Health Service component increased by 4.5 times in 3 years. Similarly, the Healthy and Safe School Environment component increased from 8.82% to 13.37% and finally to 67.98%, respectively, by years. The proportion of schools with a Healthy and Safe School Environment component increased by 8 times in 3 years. The proportion of schools with a complete Healthy Nutrition component increased from 15.29% in 2017–2018 to 30.48% in 2018–2019 and 67.98% in 2019–2020. The number of schools with full studies carried out under the title of Healthy Nutrition increased by 4.5 times in 3 years. Finally, the proportion of schools with all three components increased from 10.0% to 65.1%.

From the programs implemented to improve school health, the importance of preparing modules that include the management of common risks according to the region and local needs, and the pre-graduate and in-service training of service providers can be seen. It is thought that more successful results can be achieved by ensuring that school health management teams take an active role in these processes. Similar to these criteria, there was an increase in the number of schools fulfilling the White Flag and Nutrition Friendly School criteria. It is believed that the active role of school health management teams in these areas contributed to this increase. 

This study highlights the importance of preparing modules that include the management of common health risks according to local needs and the pre-graduate and in-service training of service providers. It is believed that by ensuring the active role of school health management teams, more successful results can be achieved. 

### 4.1. Integrating the Program into the System

In this study, in which the operational research method was used, the “Health Risk Management Program in Schools”, which was prepared after the results, was seen to be effective, and was integrated into the current SHPIP as a support in line with the decisions taken by the District School Health Board and District Hygiene Board and presented for the use of school health management teams. Since the circulation of the personnel assigned in the schools varies, the training determined in cooperation with the National Education Directorate will be implemented for new members of the school health management team or those who could not attend the trainings before. The teams of the District Health Directorate and the Directorate of National Education will continue to work in cooperation so that the school health management teams can recognize the health risks in their own schools and make the necessary interventions and necessary guidance. In order to continue this, District School Health Board and District Hygiene Board decisions were taken.

### 4.2. Limitations of the Study

Towards the end of this study, the COVID-19 pandemic, which affected the whole world, started and affected the school environment and school health studies as it affected every field. Although most of the school health studies were completed in schools before the time of the pandemic effect, the fact that schools were closed for a while and distance education came to the fore made it difficult to carry out some of the studies in the school environment.

The limitations of this study as described in the findings are valid concerns. The impact of the COVID-19 pandemic on the study, as well as the school environment, makes it difficult to fully assess the effectiveness of school health programs. The absence of students and parents as participants in the study also means that the perspectives and experiences of these important stakeholders were not taken into account. The limitation of the study design also includes the fact that only school staff members of the health management teams were targeted and included in the process, which could limit the generalizability of the findings to other countries and other parts of Türkiye. The lack of focus on a specific age range of youth, such as preschool children, primary school children or secondary school children, could also limit the applicability of the findings to specific populations. 

In conclusion, these limitations highlight the need for further research in the field of school health programs, particularly in light of the ongoing COVID-19 pandemic and the need to address the perspectives and experiences of all stakeholders, including students and parents. A more focused approach to specific age ranges of youth could also provide more detailed insights into the needs and experiences of different populations.

## 5. Conclusions

To summarize the practical implications of our findings, the success of school health programs depends on collaboration among various stakeholders, including school administration, staff, students, families and health workers. To make these programs more accessible and inclusive, it is important to have health workers on staff and to provide pre-graduate education that emphasizes risk management in school health services. Additionally, engaging students through health clubs and involving parents in parent academies and activities can also play a key role in the success of these programs. It is recommended to tailor these programs to local needs, taking into account regional differences.

## Figures and Tables

**Table 1 ijerph-20-03718-t001:** Comparison of the Evaluation Forms of the Health Protection and Improvement Program in Schools (SHPIP) by Years.

	2017–2018 ^1^		2018–2019 ^2^	
	Number (n)	Percent ^4^ (%)	Number (n)	Percent ^4^ (%)
Number of schools with full Health Service component	30	17.6	61	31.9
Number of schools with full Healthy and Safe school component	15	8.8	28	14.6
Number of schools with full Healthy Nutrition component	26	15.2	58	30.3
Number of schools with full three components	9	5.2	16	8.3
Schools have Nutrition Friendly Certificate	1	0.5	17	8.9
Schools have White Flag Certificate	2	1.1	99	51.8
Schools were not ready	105	61.7	78	40.8
Schools were ready	65	38.2	113	59.1
Total ^3^	170		191	

^1^ There were 170 schools in the district in 2017–2018. ^2^ There were 191 schools in the district in 2018–2019. ^3^ The sum of the schools recorded and evaluated gives the total number and percentage (100%) of the schools. ^4^ Percentages were given in the total number of schools on the relevant date.

**Table 2 ijerph-20-03718-t002:** 2018–2019 Evaluation Form Results.

Health Risks at School	Risk Management Areas	Answer YES (n, %) *	Answer NO (n, %) *
**School Accidents, Emergencies, Violence**	There is a first aid cabinet in the school/institution.	48 (92.3%)	4 (7.7%)
	Telephone numbers to be reached in case of emergency are posted where students and school staff can see them.	70 (84.3%)	13 (15.7%)
	There is a sufficient number of personnel who have received basic first aid certificates.	9 (10.9%)	74 (89.1%)
**Tobacco, Alcohol and Substance Use**	Students who use tobacco and/or other addictive substances, or who are thought to be, are provided to meet with the guidance teacher.	40 (100%)	0 (0%)
	School/institution employees do not use tobacco products outside the school, in areas around the school where students can see and be affected.	78 (91.7%)	7 (8.2%)
**Environmental Health Risks**	In the school health plan, goals and targets including healthy and safe school environment services have been determined.	80 (94.1%)	5 (5.9%)
	The inside and outside of the school/institution and its garden are cleaned regularly and records are kept.	70 (82.3%)	15 (17.6%)
	Toilets are regularly cleaned and records are kept.	70 (82.3%)	15 (17.7%)
	There is liquid/foam soap, dustbin and garbage bag in the bucket in the common area of the toilet.	76 (90.4%)	8 (9.5%)
	Suitable tools and materials are available for cleaning and are kept out of reach of students.	73 (85.8%)	12 (14.1%)
**Unhealthy Diet and Sedentary Life**	The cafeteria services in schools/institutions and hostels are inspected by the school administration at least once a month within the scope of the current legislation and support is received from the Provincial/District Directorate of Food, Agriculture and Livestock when necessary.	15 (71.4%)	6 (28.6%)
	In the school/institution canteen/cooperative, products such as milk and dairy products (milk, ayran, yoghurt) and/or fruit/vegetables (grain and fruit/vegetable or freshly squeezed fruit/vegetable juice) are sold.	20 (58.9%)	14 (41.1%)
	There are no advertisements, promotions, promotional posters, poster brochures encouraging the consumption of foodstuffs that may cause inadequate and unbalanced nutrition, and these products are not sold.	21 (45.7%)	25 (54.3%)
	The canteen/cooperative environments in the school/institution comply with the regulations regarding the food and beverage and the activities of the canteen/cooperative service providers.	20 (42.6%)	27 (57.4%)

* Schools in which the questions in the evaluation forms are excluded from the evaluation are not included in the total number of schools, and the percentage is calculated based on the total number of schools subject to evaluation. For this reason, the total number of schools varies for each question.

**Table 3 ijerph-20-03718-t003:** Distribution of Responses of School Staff to Items Related to Risk Management in School Health Service Areas.

School Health Service Areas	Field-Specific Risk Management Clauses	Answer YES (n, %)	Answer NO (n, %)
**School Accidents, Emergencies, Violence**	The status of accident prevention and first aid activities within the scope of school health services	191 (38.7%)	303 (61.3%)
	The status of school employees having a first aid certificate	133 (24.0%)	421 (76.0%)
	Does your school have a written standard precautionary plan on what to do in case of injury or contagious diseases?	133 (33.7%)	262 (66.3%)
	Is there a written preparation, response plan, rescue and response plan in your school that explains what to do in case of earthquake or fire?	369 (74.1%)	129 (25.9%)
	Are all employees trained and prepared for unexpected accidents, violence and suicide cases in your school?	163 (37.1%)	276 (62.9%)
	Are undesired injury and violence prevention topics covered in health education at school?	205 (43.0%)	272 (57.0%)
	Does your school prepare an emergency health response plan and inform all personnel (approach to situations such as injury, asthma attack, anaphylaxis, allergy, etc., intervention plan, process management)?	113 (26.7%)	310 (73.3%)
	Is it provided to identify and monitor the conditions of students with acute and chronic diseases?	168 (37.6%)	279 (62.4%)
**Contagious Diseases**	Effective fight against infectious diseases within the scope of school health services	85 (19.4%)	354 (80.6%)
**Tobacco, Alcohol and Substance Use**	Are the necessary visual warnings and prohibitive measures taken in your school regarding the prohibition of tobacco, alcohol and drug use?	302 (59.4%)	206 (40.6%)
	Can a ban on tobacco, alcohol and drugs be enforced for all employees in your school? Are there any information and guidance about harms and treatment?	188 (39.7%)	285 (60.3%)
	Are basic issues related to the prevention of tobacco, alcohol and drug use addressed within the scope of school health education?	283 (56.3%)	220 (43.7%)
**Environmental Health Risks**	Presence of garbage control within the scope of school health services	341(68.8%)	155 (31.3%)
	Presence of water control within the scope of school health services	200 (46.3%)	232 (53.7%)
	Providing a safe physical environment in your school, making necessary assignments (Visuals explaining safety procedures, measures to prevent falling and slipping, adequate lighting, garbage control, etc.)	335 (64.4%)	185 (35.6%)
	Are analysis tests for drinking and potable water resources carried out at your school on a regular basis?	120 (30.0%)	280 (70.0%)
	Does your school identify the source of injuries, assess the possibility of injury caused by items, and plan corrective actions?	311 (61.5%)	195 (38.5%)
	Are cleaning and maintenance services related to the entire building and its surroundings in your school provided completely and at regular intervals?	362 (68.6%)	166 (31.4%)
	Does your school ensure that the garbage collection system and pest control method are determined and implemented successfully?	342 (68.3%)	159 (31.7%)
**Unhealthy Diet and Sedentary Life**	Status of canteen inspections within the scope of school health services	380 (74.1%)	133 (25.9%)
	Do all the food and beverages sold at the school comply with the canteen regulations?	269 (55.9%)	212 (44.1%)
	Does your school address key issues related to physical activity?	247 (50.0%)	247 (50.0%)
	Does the health education in your school address the basic issues related to healthy eating?	285 (56.7%)	218 (43.3%)
	Does your school address basic issues related to chronic diseases (such as diabetes, asthma, food allergy, epilepsy)?	217 (44.7%)	269 (55.3%)
	Does your school have a written document that determines the goals and objectives of healthy nutrition services and is accessible to everyone?	165 (37.9%)	270 (62.1%)
	Does your school encourage students to have breakfast?	311 (60.2%)	206 (39.8%)
	Does your school provide a clean, safe, comfortable cafeteria/dining hall?	256 (48.7%)	270 (51.3%)
	Does your school provide food safety training to staff, especially those who are involved in catering/canteen work?	195 (48.4%)	208 (51.6%)
	Are arrangements and trainings provided for food-related emergencies (choking, food allergy, food poisoning, etc.)?	123 (28.4%)	310 (71.6%)
	Is the school canteen inspected at least once a month in accordance with the principles specified in the relevant circular?	292 (66.1%)	150 (33.9%)
**Mental Health and Behavioral Issues**	The state of conducting mental health studies within the scope of school health services	417 (78.7%)	113 (21.3%)
	Counseling, psychological and social services provided by the full-time guidance service at school	183 (37.6%)	304 (62.4%)
	Does your school carry out activities to ensure that students have access to psychological and social service support, to keep records and to include families in the process when necessary?	386 (73.9%)	136 (26.1%)
	Are students who need emotional, behavioral and mental health support in your school identified and provided with the necessary support?	371 (70.5%)	155 (29.5%)
	Are studies carried out in your school to identify students who are involved in or exposed to violence (fights, child abuse, sexual assault, etc.) and to direct them to the necessary centers?	364 (71.0%)	149 (29.0%)
	Does your school cooperate with guidance counselors and psychologists to encourage students’ social and emotional learning?	347 (69.7%)	151 (30.3%)
	Is there any work in your school to prevent harassment, violence and bullying (informing about these issues, running disciplinary mechanisms, cooperation with parents, etc.)?	353 (69.2%)	157 (30.8%)
	Are there activities in your school that will draw the attention of all students to social problems (homeless, refugee problem, disabled people, etc.) and actively be a part of the solution?	258 (52.7%)	232 (47.3%)

The answers to the survey questions were collected as “Yes” “No” “I don’t know”, and the answers to “I don’t know” were not included in the total, and the percentage calculation was based on “Yes” and “No” answers. For this reason, the total number of answers for each question varies.

**Table 4 ijerph-20-03718-t004:** Passing the First Aid Exam by Sociodemographic Characteristics.

		Status		
		Failed	Passed	*p*
		n (%)	n (%)	
**Gender (n:304)**	**Male**	9 (9.6%)	85 (90.4%)	p: 0.109 ^1^
	Female	10 (4.8%)	200 (95.2%)	
School Type (n:302)	Public	13 (5.6%)	219 (94.4%)	p: 0.401 ^2^
	Private	6 (8.6%)	64 (91.4%)	
School Level (n:302)	Special Education Vocational School	0 (0%)	7 (100%)	p: 0.027 ^2^
	High school	4 (5.7%)	66 (94.3%)	
	Kindergarten	5 (10.4%)	43 (89.6%)	
	Secondary education	1 (1.3%)	79 (98.8%)	
	Primary education	6 (7.0%)	80 (93.0%)	
	Other	3 (27.3%)	8 (72.7%)	
Position in the Institution (n:302)	Teacher	8 (4.2%)	184 (95.8%)	p: 0.013 ^2^
	Manager	7 (7.5%)	86 (92.5%)	
	Other	4 (23.5%)	13 (76.5%)	

^1^ Pearson Chi-Square Test ^2^ Fisher’s Exact Test.

**Table 5 ijerph-20-03718-t005:** Participants’ Pre and Posttest Scores.

	Pretest Score	Posttest Score	*p*
Total Score Average	69.7 ± 10.8 (Median: 72.0; Min: 20, Max: 88)	84.0 ± 10.6 (Median: 88.0; Min: 36, Max: 100)	*p* < 0.001 ^1^

^1^ Wilcoxon Test.

**Table 6 ijerph-20-03718-t006:** Comparison of Pre and Posttest Correct Answer Numbers According to Module Titles.

Module Titles	Number of Correct Answers in Pretest	Number of Correct Answers in Posttest	*p* ^1^
1. School Health (n: 117)	3.40 ± 1.12 (Min:0, Max: 5)	4.17 ± 0.92 (Min: 1, Max: 5)	*p* < 0.001
2. Healthy Nutrition, Physical Activity, Chronic Diseases (n: 117)	2.81 ± 0.86 (Min: 1, Max: 5)	3.89 ± 0.82 (Min: 2, Max: 5)	*p* < 0.001
3. Addiction (n: 117)	4.20 ± 0.95 (Min: 0, Max: 5)	4.38 ± 0.88 (Min: 1, Max: 5)	*P* < 0.036
4. Infectious Diseases, Environmental Health Hygiene (n: 117)	2.63 ± 1.17 (Min: 0, Max: 5)	3.88 ± 1.03 (Min: 1, Max: 5)	*p* < 0.001
5. Mental Health (n: 117)	4.37 ± 0.87 (Min: 0, Max: 5)	4.69 ± 0.66 (Min: 1, Max: 5)	*p* < 0.001

^1^ Wilcoxon Test.

**Table 7 ijerph-20-03718-t007:** Comparison of SHPIP Components of Schools Evaluated in 2018–2019/2019–2020 Academic Years.

SHPIP Components	2018–2019, n(%) *	2019–2020, n(%) *	*p*
**Health Service Component**			
Complete	55 (52.3%)	116 (90.0%)	*p* < 0.001 ^1^
Incomplete	50 (47.7%)	13 (10.0%)	
**Healthy and Safe Environmental Component**			
Complete	24 (22.8%)	110 (85.2%)	*p* < 0.001 ^1^
Incomplete	81 (77.1%)	19 (14.8%)	
**Healthy Nutrition Component**			
Complete	56 (53.3%)	128 (99.2%)	*p* < 0.001 ^1^
Incomplete	49 (46.7%)	1 (0.8%)	
**All Three Components**			
Complete	16 (10.0%)	105 (65.6%)	*p* < 0.001 ^1^
Incomplete	144 (90.0%)	55 (34.3%)	
**White Flag Certificate**			
Have	94 (58.7%)	126 (78.8%)	*p* < 0.001 ^1^
Do not Have	66 (41.2%)	34 (21.2%)	
**Nutrition Friendly Certificate**			
Have	16 (11.1%)	23 (14.3%)	*p* < 0.016 ^1^
Do not Have	144 (90.0%)	137 (85.7%)	

^1^ McNemar Test. * The comparison of OSKGP components was made out of 105 schools that were prepared and evaluated in the 2018–2019 academic year, and 129 schools that were prepared and evaluated in the 2019–2020 academic year, and the schools with a report were not included in the total.

## Data Availability

All datasets and analysis used throughout the study are available from the corresponding author upon reasonable request.

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
