# Peer review of "Health Risks Management Program in Schools: An Operational Study in Türkiye"

_ijerph, 2023, doi:10.3390/ijerph20043718_

Round 1
Reviewer 1 Report
This is a well written paper that addresses school health in Turkey. The study used a mixed methods design where quantitative data were collected as well as qualitative data. A large number of schools and school employees participated win the surveys which provide valuable information about the preparedness of school personnel to address school health. I do not have any major comments for the authors to consider.
The spelling of “Turkey” should be standardized throughout the paper. In some cases, it is “Turkey” and “Türkiye”.
You mention that the COVID-19 pandemic impacted data collection. Could you expand on this a little more and consider the ways that it did impact your study and the way that it could have biased the results in the later years of data collection? Could any of the improvement seen be attributed to the attention that COVID-19 placed focusing on school health in general?
Author Response
Dear Reviewer,
Thank you for your valuable recommendations. Please see the attachment for our answers.
Sincerely,
Muhammed.

Reviewer 2 Report
I would like to thank the authors for the opportunity to review this manuscript. It is well written and covers an under-researched and not extensively well-known topic. The authors do a good job at examining the current SHPIP program and then using this to form intervention strategies. However, there was insufficient detail provided in the methods section and further critical appraisal is needed to enhance the discussion and highlight the implications of this research. Specific suggestions for improvement are provided below to strengthen this manuscript for publishing.
-Abstract (page 1, line 28): analyzes should be spelled as "analyses"
-Intro (page 2, line 64): change spelling of "turkiye" to 'turkey" to be consistent with how it is spelled throughout the rest of the manuscript
-Intro: the authors have nicely summarised the SHPIP program and the need for school health programs/services/projects to be evaluated. This section could be improved by calling greater attention the "gap" or problem that the authors are trying to address and in particular, consider how this may be applicable beyond the context of Turkey. In its current form, the Introduction is written in quite a niche and specific way that focuses on heavily Turkish school health and it would be good to link this to school health more broadly.
-Introduction (page 2, lines 53-55): I agree with the authors in that adolescence is a big transitional life stage and can lead to the development of risky behaviours. The authors should also have a few sentences where they showcase the need to focus on addressing the development and health needs unique to preschool children and primary school-aged children too given that this publication targets youth aged 5-19. This is quite a broad age range and there are differing health needs and behaviours between young children (e.g. 5) and 18 or 19 year old late adolescents/young adults. This should be covered in greater detail and acknowledged in the intro.
-Methods (page 2, line 89): The authors should make clear what they mean by 'operational' study. A brief evidence-based definition would be a welcome addition here.
-Methods (page 3, line 97): use the english spelling for Turkey instead of "turkiye"
-Methods (page, 3, lines 114, 115): this sentence should be written in past tense instead of present tense
-Methods (page 4): In table 2, the % comes before the actual percentage figures. The order of this should be changed. For example, instead having (92.3%) yes for first aid cabinet.
-Methods (page 5, line 142): the authors should mention the use of quantitative research given that later on it is specified that questionnaires were used for data collection and also given that it was mentioned in the abstract
-Methods (page 5, line 152): can the authors comment on the reliability and validity of this SHI index that was used?
-Methods (page 5, line 161): was this a purpose-designed study? Can the authors discuss the reliability of this measure?
-Methods (page 5, lines 167-175): More detail is needed concerning the focus group. How many people participated in the focus group? Can example questions be provided in addition to specifying the topics were discussed or could the interview questions be included as an Appendix? Additionally, the authors mention that content analysis was performed, which is different from what was stated in the abstract (phenomenological analysis method stated on page 1, line 20). Who facilitated the focus groups/ Were the trained in qualitative interviewing? Were participants compensated for their time? Were participants given the opportunity to review the transcripts. It is strongly advised that the authors used the Consolidated Criteria for Reporting Qualitative Research (COREQ) checklist by Tong et al. (2007) to report on the qualitative methods used.
-Methods (page 5, lines 176-178): There is insufficient detail regarding how the questionnaire data were analysed (eg what statistical tests were used? just descriptive statistics? etc.), as well as how the qualitative data were analysed. For example, was NVivo used for analysing the focus groups?
-Results (page 9, lines 236-241): the results from the focus groups are very brief. This section could be improved by having some example quotes and going into a bit of greater detail concerning the key themes.
-Discussion (page 12, lines 333-335): This information would be good to tie into the introduction to highlight the gap that this paper is addressing in the literature
-Discussion (page 12, lines 337): the authors refer to the use of a mixed method design here, yet earlier other terms were used. consistency in use of methodological terms is needed
-Discussion: The authors do a nice job at summarising key findings but the link of the findings to existing literature is quite basic level. Greater critical appraisal and delving deeper into the link between the findings and previous studies would improve this section
-Limitations: The authors could also comment on the disadvantages of the study design and lack of generalisability to other countries and other parts of Turkey. Another limitation would be the broad age range of youth as opposed to focusing in greater detail on preschool children, primary school children or secondary school children
-Discussion and conclusions: the authors could further improve their focus on the practical implications of their findings
Author Response
Dear Reviewer,
Thank you for your valuable recommendations. Please see the attachment for our answers.
Sincerely,

Round 2
Reviewer 2 Report
I commend the authors on revising the manuscript to address all key points noted. The Introduction now better highlights the gap in the literature and the Methods now have sufficient detail, especially related to the qualitative component of this paper. The Discussion also now better acknowledges the limitations. Well done!